# Poly-ligand profiling differentiates trastuzumab-treated breast cancer patients according to their outcomes

Valeriy Domenyuk[1], Zoran Gatalica[1], Radhika Santhanam[1], Xixi Wei[1], Adam Stark[1], Patrick Kennedy[1], Brandon Toussaint[1], Symon Levenberg[1], Jie Wang[1], Nianqing Xiao[1], Richard Greil[2], Gabriel Rinnerthaler[2], Simon P. Gampenrieder[2], Amy B. Heimberger[3], Donald A. Berry[4], Anna Barker[5], John Quackenbush[6], John L. Marshall[7], George Poste[1,5], Jeffrey L. Vacirca[8], Gregory A. Vidal[9], Lee S. Schwartzberg[9], David D. Halbert[1], Andreas Voss[1], Daniel Magee[1], Mark R. Miglarese[1], Michael Famulok[1,10,11,12], Günter Mayer[1,10,12] & David Spetzler[1]

Assessing the phenotypic diversity underlying tumour progression requires the identification of variations in the respective molecular interaction networks. Here we report proof-of-concept for a platform called poly-ligand profiling (PLP) that surveys these system states and distinguishes breast cancer patients who did or did not derive benefit from trastuzumab. We perform tissue-SELEX on breast cancer specimens to enrich single-stranded DNA (ssDNA) libraries that preferentially interact with molecular components associated with the two clinical phenotypes. Testing of independent sample sets verifies the ability of PLP to classify trastuzumab-treated patients according to their clinical outcomes with ROC-AUC of 0.78. Standard HER2 testing of the same patients gives a ROC-AUC of 0.47. Kaplan–Meier analysis reveals a median increase in benefit from trastuzumab-containing treatments of 300 days for PLP-positive compared to PLP-negative patients. If prospectively validated, PLP may increase success rates in precision oncology and clinical trials, thus improving both patient care and drug development.

---

[1] Caris Life Sciences, 4610 South 44th Place, Phoenix, AZ 85040, USA. [2] IIIrd Medical Department, Oncologic Center, Paracelsus Medical University Salzburg, Austria and Salzburg Cancer Research Institute, and Cancer Cluster Salzburg, Müllner Hauptstraße, 48A-5020 Salzburg, Austria. [3] Department of Neurosurgery, University of Texas MD Anderson Cancer Center, 1400 Holcombe Boulevard, Houston, TX 77030, USA. [4] Berry Consultants, LLC, 3345 Bee Caves Road, Suite 201, Austin, TX 78746, USA. [5] Complex Adaptive Systems Initiative, Arizona State University, 1475N. Scottsdale Road, Suite 361, Scottsdale, AZ 85257, USA. [6] Department of Biostatistics and Computational Biology, Dana-Farber Cancer Institute, 450 Brookline Avenue, Boston, MA 02215-5450, USA. [7] Department of Oncology, Lombardi Comprehensive Cancer Center, Georgetown University Medical Center, 3800 Reservoir Road, N.W., Washington, DC 20057, USA. [8] North Shore Hematology Oncology Associates Cancer Center, 226N Belle Mead Road, East Setauket, NY 11733, USA. [9] Division of Hematology and Oncology, University of Tennessee Health Science Center, 7945 Wolf River Boulevard, Memphis, TN 38138, USA. [10] LIMES Program Unit Chemical Biology & Medicinal Chemistry, c/o Kekulé Institute for Organic Chemistry and Biochemistry, University of Bonn, Gerhard-Domagk-Straße 1, 53121 Bonn, Germany. [11] Chemical Biology Max-Planck-Fellowship Group, Center of Advanced European Studies and Research (CAESAR), Ludwig-Erhard-Allee 2, 53175 Bonn, Germany. [12] Center of Aptamer Research and Development, University of Bonn, Gerhard-Domagk-Straße 1, 53121 Bonn, Germany. Correspondence and requests for materials should be addressed to M.F. (email: m.famulok@uni-bonn.de) or to G.M. (email: gmayer@uni-bonn.de) or to D.S. (email: dspetzler@carisls.com)

Precision oncology emanates from the premise that clinical outcomes will be improved by the selection of treatment regimens that act on specific molecular targets identified by multi-Omics profiling. Although this premise has been validated in several clinical studies[1–3] and provides a rationale for ongoing trials, such as TAPUR (NCT02693535), there are still troubling examples of discordance in which patients positive for the deemed target(s) for drug action fail to respond and patients presumably lacking the target(s) respond[4,5]. This problem is epitomized by the clinical dilemma in the response of metastatic breast cancer patients to the HER2 antagonist trastuzumab (Herceptin™), alone or in combination, from which 38–50% of HER2-positive patients derive no clinical benefit[6–9] (U.S. BL 103792 Supplement: Trastuzumab-Genentech, Inc. 103791 of 103732/Regional (STN: BL 103792/105256): Final labelling 10292010). Conversely, there may be patients whose tumours express little to no HER2 who nonetheless derive benefit[10,11]. The high cost of many new cancer treatments, most notably the new classes of targeted agents and immune checkpoint inhibitors, along with the high proportion of patients whose diseases do not respond to these treatments, highlights the compelling clinical and economic needs for complementary and companion diagnostic tests[12] to differentiate patients who will benefit from a specific treatment regimen from those who will not[13,14].

Differences in patient responses to cancer treatment reflect subtle alterations in tumour system states[15–18]. Therefore, approaches that stratify these patients must simultaneously survey the multiplicity of molecular features that underlie the complex and largely unpredictable drug response phenotypes. A prerequisite for assessing the phenotypic diversity generated by tumour progression is the ability to identify variations in the underlying molecular interaction network or interactome[19,20]. The interactome is estimated to consist of millions of multi-molecular complexes[21]. Individual or small numbers of biomarkers are thus unlikely to adequately reflect treatment benefit[4,5,13]. Deconvolution of multi-nodal perturbations in network architectures requires developing technologies that simultaneously assess millions of biological features in an unbiased manner and consolidate the result into an unambiguous readout. Profiling this level of network diversity therefore requires an unbiased, hypothesis-free approach that must employ an equal or greater number of potential detector molecules[22].

Towards this goal, here we use tissue-SELEX on formalin-fixed paraffin-embedded (FFPE) tumour tissue samples from breast cancer patients to enrich libraries of single-stranded DNA (ssDNA) that utilize Watson–Crick base pairing, sequence-specific protein binding or aptameric binding similar to antibodies for interacting with molecular targets and protein complexes in the sample. To assess the ability of poly-ligand profiling (PLP) to differentiate breast cancer patients who did (B) or did not (NB) derive benefit from treatment with trastuzumab alone (T) or in combination with chemotherapy (C+T), we compare PLP results with immunohistochemical (IHC) analysis of HER2, which is overexpressed in approximately 20% of human breast cancers[23,24]. Testing of independent sample sets verifies the ability of PLP to differentiate patients as assessed by an area under the receiver-operating characteristic curve (ROC-AUC) value of 0.78 using 10-fold cross-validation (CV). Standard anti-HER2 IHC testing of the same patients gives a ROC-AUC of 0.47. Kaplan–Meier (KM) plots of PLP test-positive patients have a median duration of 429 days of benefit from trastuzumab-containing treatments versus 129 days for test-negative cohorts. These data demonstrate that PLP enables access to intricate molecular phenotypes that reveal underlying differences in treatment response. PLP may increase success rates in precision oncology and of pivotal drug trials in the future and thus facilitate both drug development and patient care.

## Results

**Enrichment of ssDNA libraries on FFPE tumour samples.** Traditional controlled clinical trials enroll homogenous populations of patients that often fail to faithfully represent the heterogeneous patient populations typically treated in community practice[25]. To reflect bona fide clinical practice, we designed our study retrospectively by using patient samples routinely submitted to certified central laboratories. Because our study was not a prospective trial, we employed time to next therapy (TTNT) as a surrogate measure of treatment efficacy. TTNT is defined as the time that elapsed between the initiation of trastuzumab-based therapy and the initiation of a subsequent line of therapy[26]. Samples from patients with TTNT <6 months represent the no benefit (NB) category for library enrichment, whereas the TTNT of samples from the benefit (B) category was >12 months. Comprehensive clinical information for each metastatic breast cancer patient included in this process is provided in Supplementary Data 1. All tissue samples used for enrichment and testing were collected prior to treatment to ensure that library enrichment is independent of molecular changes in the tumour tissue resulting from treatment (Fig. 1, Methods, Supplementary Data 2).

A starting library of $10^{13}$ unique ssDNAs was subjected to several rounds of positive and negative selection[27,28] in situ to identify sub-libraries that preferentially bound to FFPE tumour tissue sections from breast cancer patients who did or did not derive benefit from C+T regimens. We refer to this process as library enrichment. For example, to generate libraries preferentially binding NB cases the starting library was subjected to positive selection using one NB case (rounds 1–6) and negative selection using two B cases (rounds 4–6) (Fig. 2 and Methods). An opposite selection logic was employed to enrich libraries with preferential B-case binding. In total, 9 B cases and 8 NB patient samples were used for enriching the 17 different libraries (Supplementary Data 1). The two best-performing enriched libraries (EL) for NB patients (EL-NB) and for B patients (EL-B) (Fig. 1, $n = 6$, enrichment and Methods) were selected for further testing based upon the reduction of staining on the negative enrichment samples, accumulation of staining compared to unenriched library and differential staining on the other independent enrichment samples (Fig. 3).

Specifically, EL-NB was obtained by incubating the starting library directly with FFPE-fixed breast tumour tissue from a C+T-treated NB patient (Fig. 2a, Methods). After 1 h of incubation (step i), non-binding ssDNAs were removed by washing (step ii), followed by a Nuclear Fast Red (NFR) staining and microdissection of the tumour tissue away from normal adjacent tissue (step iii) and asymmetric PCR amplification of ssDNAs bound to the dissected tumour tissue. These positive selection steps were performed three times on serial tissue sections of the same case (rounds 1–3; Fig. 2a, c). The ssDNA library from the third round of enrichment was then subjected to two consecutive counter-selection steps by incubating with FFPE tissue from two C+T-treated B patients. The supernatant from the second counter-selection step, i.e., the library that is depleted of ssDNAs associated with binding to B tumour tissue, was transferred to a new tissue section from the original NB patient's tumour for one final positive selection step as described above and the subset of bound ssDNAs was PCR-amplified (Fig. 2b). The negative–negative–positive steps were performed a total of three times (SELEX rounds 4–6; Fig. 2c).

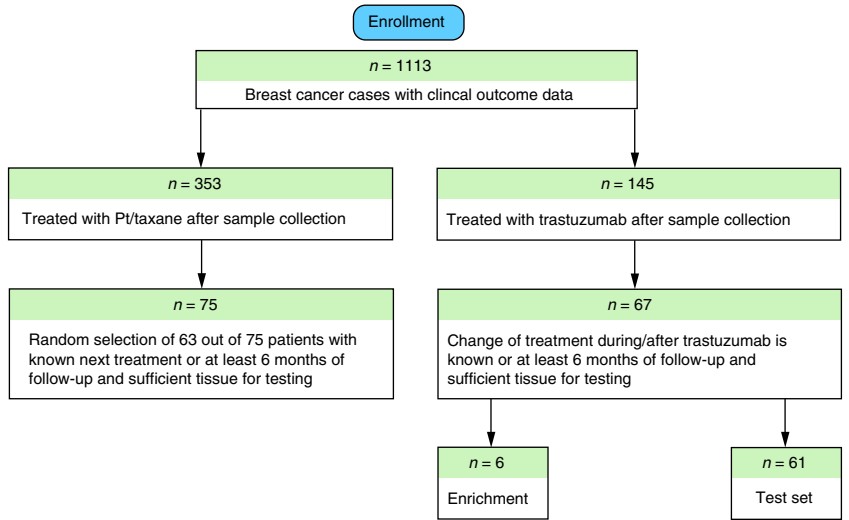

**Fig. 1** Enrollment of patients for poly-ligand profiling. The flow chart illustrates the rationale and the numbers of patients for inclusion in the test set and those used for the ssDNA library enrichment. A comparative analysis of the demographic and clinical details for the excluded 78 patients and the enrolled 67 patients (Supplementary Data 2) shows no bias among tested clinical and demographic factors, thus the 67 enrolled cases are representative

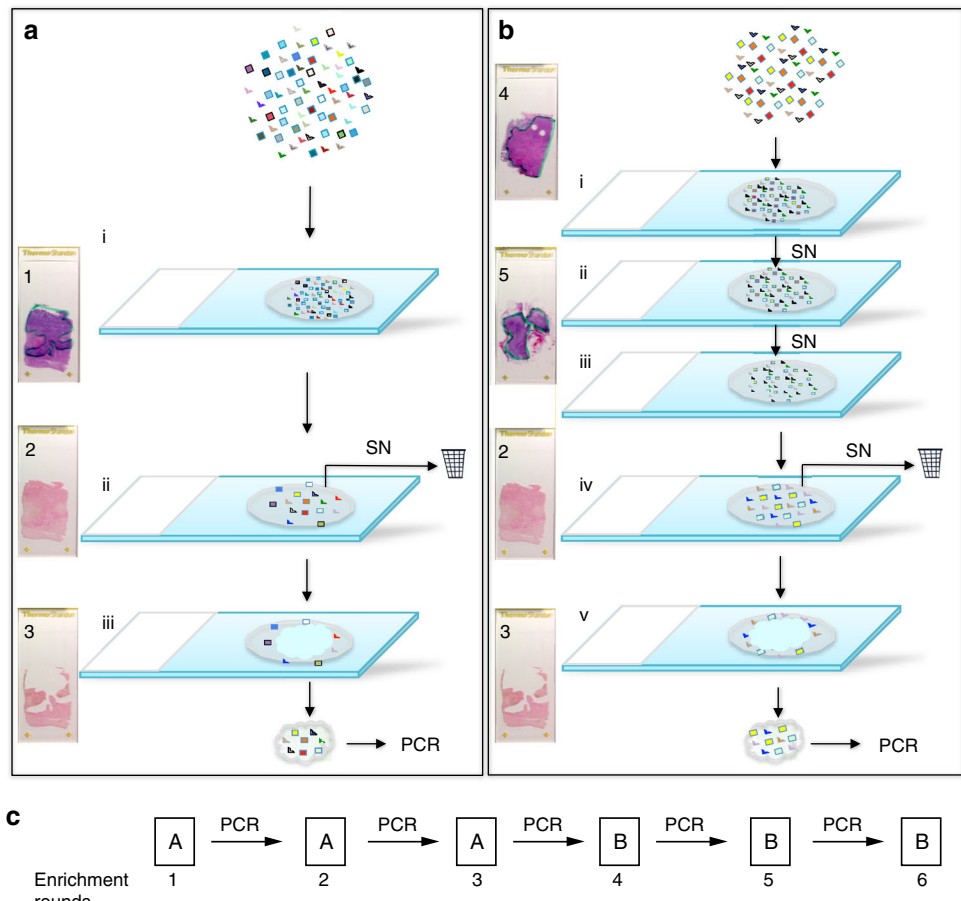

**Fig. 2** ssDNA library enrichment. **a** Positive enrichment steps towards a library that identifies non-benefitting (NB) cases: (i) incubation of the ssDNA library with the NB tissue; (ii) removal of unbound sequences, (iii) dissection of tumour tissue and recovery of the subset of sequences, specific to the NB cancer tissue. SN, supernatant. Recovered ssDNAs were amplified by PCR, converted to ssDNAs and used for the next enrichment round. Slide images on the left show tissue appearance: Slide (1) Haematoxylin and eosin (H&E) staining of NB tissue (tumour area outlined in green); Slide (2) Nuclear Fast Red (NFR)-stained tissue after library partitioning before dissection; Slide (3) Remaining normal tissue after dissection of cancer tissue with bound ssDNAs. **b** Enrichment steps with additional counter-selection steps on benefit (B) cases: (i) incubation of the ssDNA library with the first B tissue; (ii) incubation of the supernatant from (i) with the second B case; (iii) incubation of the supernatant from (ii) with the NB case from **a**; (iv) and (v) correspond to the steps (ii) and (iii) in **a**, respectively. Slide images on the left show tissue appearance: Slide (4) H&E staining of first B tissue (tumour area outlined in green); Slide (5) H&E staining of second B tissue (tumour area outlined in green); Slides 2 and 3 are same as in **a**. **c** The entire ssDNA library enrichment is comprised of three enrichment rounds as shown in **a**, followed by three enrichment rounds as shown in **b**

The staining (adopted from IHC, Methods) with the round 6 library (EL-NB) of the NB case tissue, used for the enrichment, showed a significant increase of intensity (Fig. 3a, round 6), compared to no or weak staining for the unenriched library (Fig. 3a, round 0). Similar staining was observed on a separate NB case not used for the library enrichment (Fig. 3b). Not unexpectedly, staining intensity was also high when the PCR-amplified EL-NB from round 3 (EL-NB before counter selection) was applied to a B case (Fig. 3c, left panel). However, after round 5 a notable decrease in staining intensity was observed for EL-NB applied to the same B case, indicating that the counter-selection steps were effective (Fig. 3c, right panel). Thus EL-NB shows high staining intensities on NB tissue but stains the B case used for counter-selection with considerably weaker intensity. EL-NB was

therefore potentially suitable for PLP staining on an independent test set.

We carried out a separate enrichment process in the opposite direction, using B cases for positive, and NB cases for counter-selection to generate library EL-B, which showed preferential staining of B cases compared to the NB cases or the non-enriched starting library (Supplementary Fig. 1). Taken together, the staining characteristics observed with EL-NB and EL-B indicate successful enrichment towards the targeted phenotypes. Next-generation sequencing (NGS) of the enriched libraries showed a decrease in library complexity from $10^{13}$ unique ssDNAs to $1.4 \times 10^{7}$ and $8.5 \times 10^{6}$ for EL-B and EL-NB, respectively (Supplementary Data 3). Since the library input was constant in all enrichment rounds, accumulation of the sequences with high

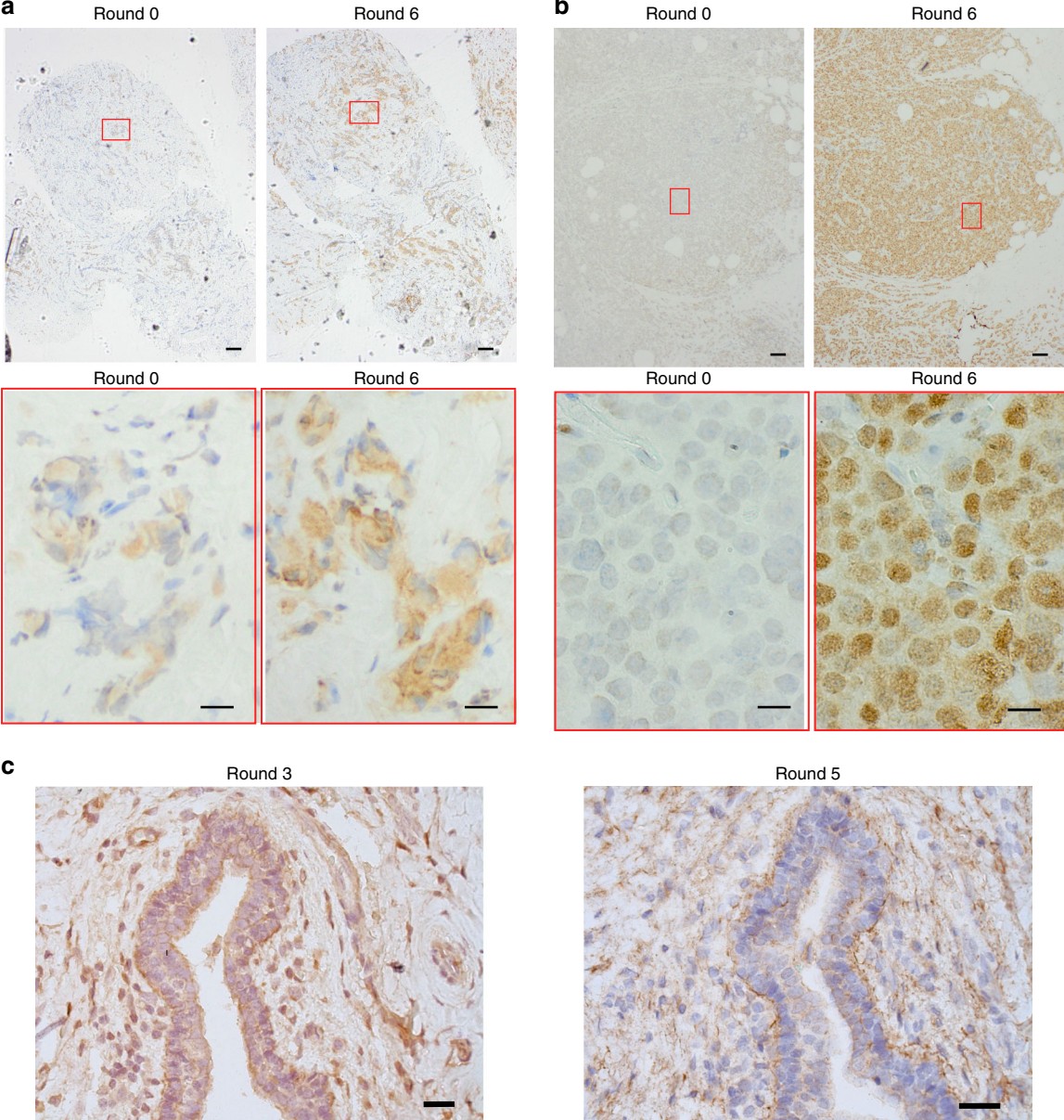

**Fig. 3** Enriched ssDNA library validation. **a** Staining of the tissue from the NB case that was used during the selection process with unenriched library (round 0), compared to the enriched EL-NB library (round 6; upper panel: ×4, 100 μm scale bar, lower panel: ×60, 14 μm scale bar); **b** Staining of tissue from an NB case not used during the selection process with unenriched library (round 0), compared to the enriched library EL-NB (round 6; upper panel: ×4, 100 μm scale bar, lower panel: ×60, 14 μm scale bar); **c** Staining of the tissue from the benefiter case employed for counter selection in the enrichment of EL-NB, using the output ssDNAs from round 3 (left, ×20, 40 μm scale bar), compared to the output ssDNAs from round 5 (right, ×20, 40 μm scale bar). No adjustment was applied to images from microscopy

counts, shown in Supplementary Fig. 2, indicates affinity maturation of the ssODN libraries during enrichment.

**Testing enriched libraries on independent patient samples**. We next tested the performance of the enriched libraries EL-B and EL-NB on a cohort of patient samples independent from those used for enrichment (Supplementary Data 4), covering a TTNT range from 49 to 1792 days, referred to as test set. For this purpose, the test set cases were stained (Methods) with both enriched libraries (Supplementary Fig. 3). We employed histological *H*-scoring as a read out of the tissue staining, similar to standard pathological practice for IHC[29]. The scoring for both cytoplasmic and nuclear staining (Fig. 4, Methods) was performed by a board-certified pathologist who was blinded to the patient outcomes (Supplementary Data 5). Examples of PLP staining intensity levels in the cytoplasm and the nuclei of breast cancer FFPE tissue ranging from 0 to 3+ are shown in Fig. 4a. The histological scores were calculated by standard methods by determining the percentage of cells on the entire tissue classified to fall within each PLP intensity level in the cytoplasm and the nucleus, respectively (Fig. 4b, Methods). To evaluate staining and scoring reproducibility, we selected cases that showed weak and

strong staining with EL-NB and then scored nuclear staining between technical replicates (Fig. 4c, upper panel) and different batches of library (Fig. 4c, lower panel). The classification of four strongly and weakly staining cases was consistent and independent of these variables, indicating that the staining and scoring is reproducible (Supplementary Data 6).

To further assess the technical reproducibility of PCR-amplified versions of EL-NB and EL-B, both libraries were amplified for up to 10 PCR generations. Each PCR generation of EL-NB and EL-B was then used for the staining of consecutive tissue sections from two unrelated breast cancer patients, namely, patient 1 as an example for relatively low and patient 2 for relatively high staining intensity (Supplementary Fig. 4A). No significant difference in the staining was observed from generation to generation, indicating highly robust performance of both libraries after multiple PCR amplification. Moreover, NGS analysis[30] of the sequence composition of EL-NB and EL-B after subsequent 9 generations of PCR amplification (Supplementary Fig. 4B) showed negligible change in the first 8 generations of library EL-B and in the first 6 generations of library EL-NB at different sequence cut-offs up to the top 10,000 sequences. As expected, in generations 7–10 (EL-NB) and 9 and 10 (EL-B) a

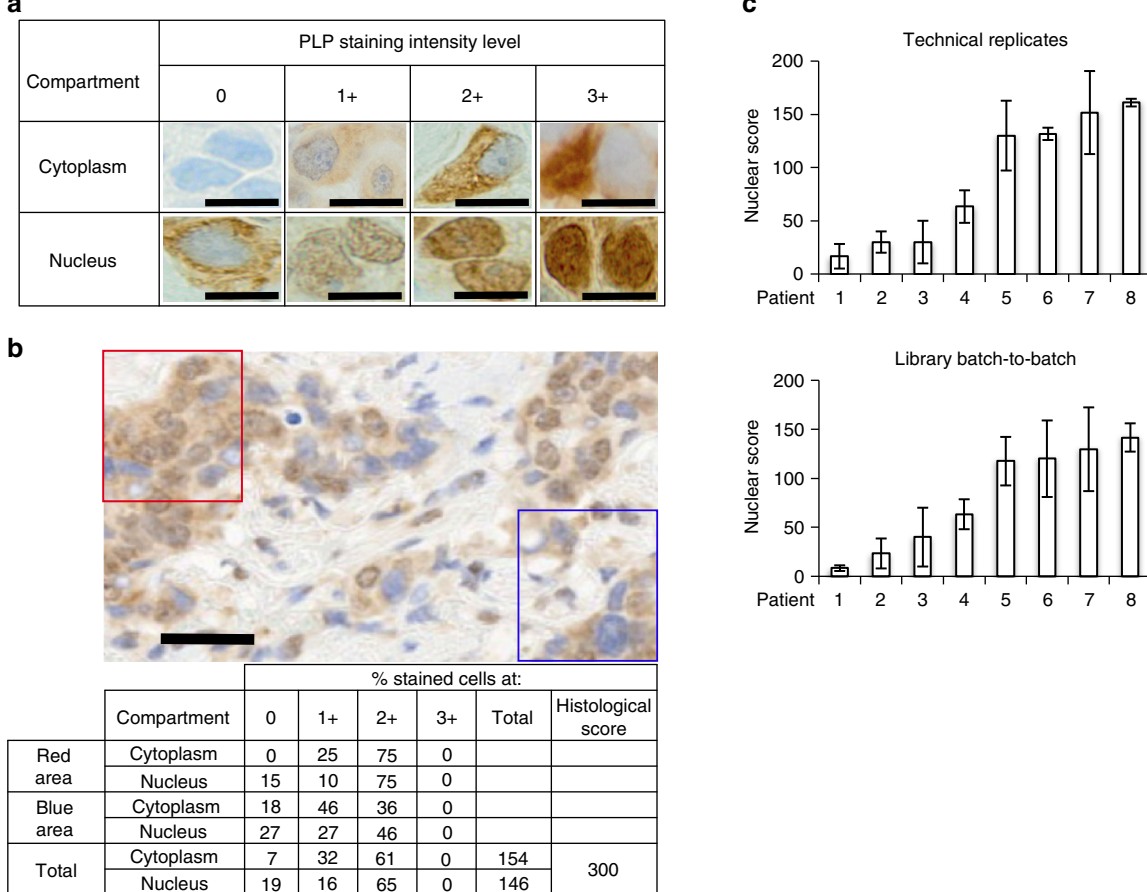

**Fig. 4** Histological score calculation based on the intensities of PLP staining and reproducibility of the staining. **a** Representative examples of the different PLP intensity levels in the cytoplasm and the nucleus of breast cancer FFPE tissue cells, compiled from different cases and different enriched libraries (×60, 10 μm scale bar). **b** Examples of the histological score calculations in two areas of the breast cancer tissue, stained with EL-NB. Note: assessment of the percentage of cancer cells with a particular staining intensity is usually performed on the entire tissue. Correspondingly, each stained case will have its unique cytoplasmic, nuclear and combined scores. Depending on the marker localization, nuclear, cytoplasmic or combined PLP scores can be informative for the differentiation of the benefiter from non-benefiter cases (×20, 20 μm scale bar). No adjustment was applied to images from microscopy. **c** Reproducibility of the PLP nuclear staining and scoring using EL-NB on eight different patients between technical replicates and different batches of the library preparation. TL-NB staining of the cytoplasm was not informative for the differentiation of B and NB cases. Patients 1–4 were selected as examples of relatively weak intensity staining, patients 5–8 are examples of relatively strong intensity staining (error bars are s.d. from three replicates)

declining trend of perfectly matching sequences is observed, most likely due to point mutants of the original sequences that slowly increase in their representation at these later generations in the library. Importantly, such changes in library composition did not compromise the staining intensity and profile (Supplementary Fig. 4A).

**Stratification of B and NB patients by PLP**. With two libraries that reciprocally show preferential staining for either B or NB cases, we hypothesized that PLP scores may be informative for the differentiation and the classification of clinical benefit and non-benefit from C+T or T regimens in the 61 independent test cases (Fig. 1, Supplementary Data 4). The PLP scores of EL-B and EL-NB (Supplementary Data 5) were assessed by ROC curves, and calculated AUC values (Fig. 5a, b). For EL-NB, an AUC value of 0.73 was obtained based on the nuclear scoring (Fig. 5a), whereas EL-B yielded an AUC value of 0.63 (Fig. 5b) based on nuclear and cytoplasmic scoring. These AUC values from the individual libraries indicate that EL-NB and EL-B are revealing consistent biological information that distinguishes the NB and B phenotypes. To classify the patient's response to C+T or T regimens with a multivariable method that uses PLP scores from both EL-NB and EL-B staining, a logistic regression model was developed. Specifically, the binary outcome of benefit or non-benefit status was used as the dependent variable. The staining scores of EL-NB and EL-B, respectively, were treated as independent variables. We then assessed the performance of the model by ROC curve analysis. By combining the data from the two libraries, the AUC value increased to 0.81, indicating improved performance due to the reciprocal nature of the enrichment schemes (Fig. 5c, blue). The statistical reliability of this analysis was further verified by 10-fold cross-validation (CV), which resulted in an AUC value of 0.78 (Fig. 5c, black). We then compared these PLP-based NB and B classifications with those predicted by HER2 IHC scoring of the same 61 cases' test set (Fig. 5c, red). The HER2 IHC results yielded an AUC value of 0.47, indicating that EL-NB and EL-B outperformed conventional HER2 IHC in classifying trastuzumab B and NB cases in this population. Importantly, EL-NB and EL-B were able to effectively classify B and NB patients defined by IHC

as either HER2-negative/equivocal (AUC = 0.73, Supplementary Fig. 5A) or HER2-positive cancers (AUC = 0.84, Supplementary Fig. 5B). Thus, although the ability of the libraries to differentiate NB from B cases is independent of HER2 (Supplementary Fig. 5A, B), the number of PLP+ patients is lower in the HER2-negative group than in the HER2-positive group, as expected (Supplementary Fig. 5C).

To determine whether the EL-NB and EL-B phenotypes were revealing information about response to C+T or T regimens and not simply classifying patients with a favourable prognosis regardless of trastuzumab-containing treatment, we stained FFPE tumour tissues from an independent cohort of 63 breast cancer patients who were treated with chemotherapy (C) without trastuzumab. Like all other samples in this study, the samples from this cohort were collected prior to treatment. The combined PLP scores for the patients treated with C resulted in an AUC value of 0.53 (Fig. 5c, purple), indicating that the performance of EL-NB and EL-B in C+T set relates to the molecular profile determining the response to the presence of trastuzumab in the treatment regimens. Moreover, since trastuzumab has been reported to be substantially less effective in oestrogen receptor (ER)-positive breast cancer[31,32], it is important to note that the ER status of all cases enrolled in our study showed no correlation with the benefit from trastuzumab-containing therapy (Supplementary Fig. 5D, Supplementary Data 4). Taken together, these data indicate that the application of these libraries to the 61 test set cases differentiates patients with benefit from patients without benefit from C+T or T treatment with unprecedented accuracy, regardless of their HER2 status (Fig. 5, Supplementary Fig. 5).

To evaluate the relationship of PLP to clinical outcomes as measured by TTNT[26], we performed a KM analysis on the 61 patients who received C+T or T. TTNT is a US Food and Drug Administration-approved clinical endpoint that is robustly captured by electronic medical record systems and therefore is generally available. We established a threshold by determining the point in the ROC curve that has the shortest distance to the point at which sensitivity and specificity were both 100% using the fitted values from the logistic regression model (Fig. 5c, blue dot). The resulting KM curves show that cases with positive PLP test results exhibited a significantly longer TTNT (Fig. 6a, blue curve)

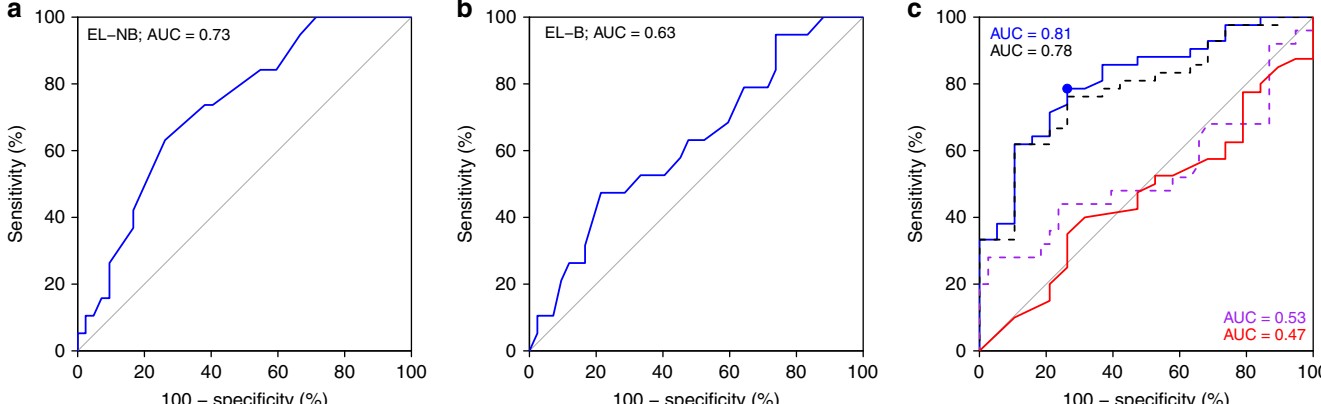

**Fig. 5** Receiver operating characteristic (ROC) curves for the differentiation of chemotherapy+trastuzumab (C+T) or trastuzumab monotherapy (T) B and NB cases using histological scores from poly-ligand profiling (PLP) with libraries EL-NB and EL-B in the test set of 61 cases. **a** ROC curve obtained based on EL-NB library PLP scores. AUC 0.73, p = 0.01. **b** ROC curve obtained based on EL-B library PLP scores. AUC 0.63, p = 0.11. **c** ROC curves obtained based on combined PLP scores of EL-NB and EL-B libraries using a logistic regression algorithm (blue curve, AUC = 0.81, p = 0.0007); black dashed curve: 10-fold cross-validation on the combined scores of EL-NB and EL-B libraries (AUC = 0.78, p = 0.002), 10-fold cross-validation was run 100 times and the average predictive score was used as the final cross-validation result; red curve: AUC values obtained by HER2-IHC scoring of the same 61 cases (AUC = 0.47); purple dashed curve: performance of the combined libraries on 63 HER2 cases independent from the test set who received platinum/taxane combination therapy instead of trastuzumab (AUC = 0.53). TTNT data for these patients are unrelated to trastuzumab-based regimens

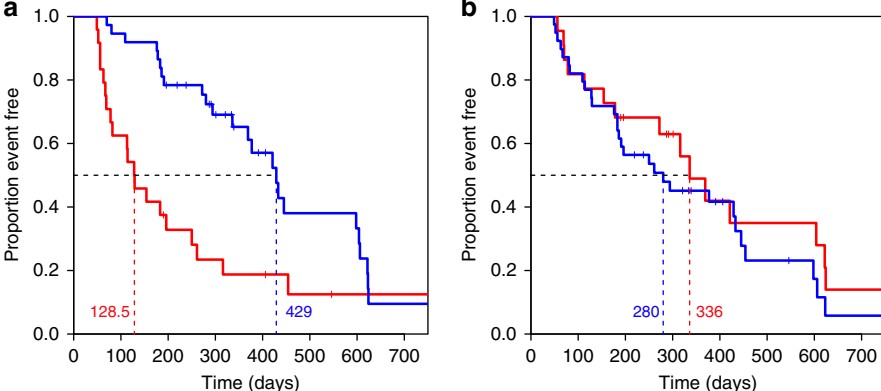

**Fig. 6** Kaplan–Meier curves of C+T- or T-treated breast cancer patients stratified by poly-ligand profiling or by tumour Her2 status. **a** Kaplan–Meier curve of C+T- or T-treated breast cancer patients stratified by poly-ligand profiling. The shortest distance between the ROC curves to point (specificity and sensitivity = 100%) determines the cutoff of test positive and negative and is represented as the blue point in Fig. 5c (Sensitivity: 78.6%; Specificity: 73.7%). "Event" is defined as the time point (days) at which a patient either deceased from cancer or at which trastuzumab-based treatment changed. Median time of benefit is 429 days for patients tested positive (blue, $n = 37$, event = 25) and 129 days for patients tested negative (red, $n = 24$, event = 20). HR = 0.384, 95% CI: 0.21–0.70; log-rank $p = 0.001$. The small vertical lines mark cases that were censored due to absence of treatment follow-up data (Supplementary Data 4). **b** Kaplan–Meier curve of C+T- or T-treated breast cancer patients stratified by tumours' HER2 status. HER2 status of patients with both HER2 IHC and HER2 ISH test results was determined according to ASCO guidelines[36]. Median time of benefit is 280 days for HER2 positive cases (blue, $n = 39$, event = 29), and 336 days for HER2-negative/equivocal cases (red; $n = 22$, event = 16). HR = 1.27, 95% CI: 0.68–2.37; log-rank $p = 0.448$

than those in the PLP-negative cohort (Fig. 6a, red curve). The median event-free time, i.e., the time that elapsed before the treatment regimens changed, increased from 129 days for the test-negative to 429 days for the test-positive cases. For comparison, we performed the same analysis using HER2 status as determined by IHC and in situ hybridization. In this case, the KM curves did not reveal any significant difference in TTNT between patients who were HER2+ (280 days) or HER2-negative/equivocal (336 days) (Fig. 6b). Importantly, a modified KM analysis on the same 61 patients using 10-fold CV (Supplementary Fig. 6A) still resulted in significant differences in the KM curves between the PLP-positive versus -negative cohorts (Supplementary Fig. 6B).

## Discussion

In conclusion, our data demonstrate the potential of PLP to improve patient stratification for eligibility for trastuzumab treatment. PLP enables access to intricate molecular phenotypes underlying variations in treatment response. A current limitation of this study is that it was performed retrospectively. Accordingly, a prospective trial is required to establish the clinical utility of PLP. In principle, the application of PLP should not be limited to breast cancer since it offers a flexible platform for rapid development of novel poly-ligand libraries that differentiate therapeutic responder and non-responder populations in diverse cancers and different therapeutic regimens. We are currently applying PLP to other cancers and various treatment modalities. Previous studies have used morphology-based enrichment of ssDNA libraries on cancer tissues[33,34] aimed at the important goal of identifying new biomarkers. However, this was explicitly not the purpose of the present study, which was to establish a proof-of-concept for poly-ligand libraries that can address the heterogeneity of molecular composition and the complex interactomes that reflect intra- and inter-tumour heterogeneity. In the future, PLP may increase success rates in precision oncology and of pivotal drug trials and thus might facilitate drug development and improve patient care. We further contend that the assay platform described here holds promise for identifying patients most likely to derive significant clinical benefit from given

treatment regimens. Equally importantly, PLP could mitigate the clinical, economic and ethical issues associated with futile therapies.

## Methods

**FFPE tissue cases**. This study was performed with Western Institutional Review Board approval, 45 CFR 46.101(b)(4). The study was performed utilizing retrospective clinical data and de-identified remnant samples. The study is considered IRB exempt.

This retrospective study included cases from women with invasive breast cancer who received C+T or T monotherapy first-line treatment after tissue collection with a sufficient number of properly fixed and embedded slides available. Cases with in situ cancer and improperly fixed or crushed tissue sections were not included in this study. Cases with incomplete staining (i.e., insufficient coverage of the tissue with binding solution) and other technical problems with the assay performance were excluded from analysis.

In total, 33 unique cases with available TTNT data from C+T therapy were used for enrichment of 17 ssDNA libraries towards B or NB cases (Supplementary Data 1). Two of these libraries, EL-NB and EL-B, were selected for further characterization on unrelated cases (test set, $n = 61$) with available TTNT data from C+T or T therapies. A separate group of 63 cases that had TTNT data from chemotherapy excluding trastuzumab ("C") (Supplementary Data 4) was used as a control set to evaluate the specificities of EL-NB and EL-B towards C+T or T based TTNT. Note, patients' enrollment (Fig. 1) is reflecting the number of patients used for enrichment and characterization of the specified EL-NB and EL-B libraries. Excised tissue containing both tumour and normal parts was FFPE and serially sectioned (4 µm). Haematoxylin and eosin (H&E) staining was performed for 1–2 slides of each case and served for initial pathological diagnosis. Pretreatment of the FFPE tissue slides before the enrichment included incubating slides at 60 °C for 1.5 h, followed by automated deparaffinization and epitope retrieval on the Ventana UltraView Autostainer. Specifically, deparaffinization at 72 °C for 24 min, dehydration by ethanol, epitope retrieval at 90 °C for 36 min and 100 °C for 4 min (pH 8), followed by peroxidase inhibition ($H_2O_2$, $1\% \leq x < 5\%$) and washing slides with detergent (Dawn 1-00; P&G Professional) to remove residual liquid coverslip. For testing of the enriched libraries, deparaffinization was performed manually, by incubation at 60 °C for 1.5 h, followed by epitope retrieval on PT-Linker (Dako) at 97 °C for 20 min, pH 9.

A subset of 1 NB breast cancer cases and 2 B breast cancer cases (or vice versa) was selected for a particular enrichment (Supplementary Data 1). For enrichment purposes, tissue areas with breast carcinoma were utilized as positive selection targets, while adjacent non-malignant tissue as well as carcinoma tissue from patients with alternative response was used as counter-selection targets.

**ssDNA library reproduction by asymmetric PCR**. The random ssDNA library was designed to accommodate one-step NGS preparation[22]. The library (sense strand) was synthesized at Integrated DNA Technologies (IDT, USA) and PCR-

amplified using biotinylated sense primer and polyA/iSp9-modified antisense primers:

5′-Biotin-CTA GCATGACTGCAGTACGT3′
5′AAAAAAAAAAAAAAAAAAAAAAAAAAAAAAAAAAAA/iSp9//iSp9/TCGTCGGCAGCGTCA3′
5′CTAGCATGACTGCAGTACGT-35N-CTGTCTCTTATACACATCTGACGCTGCCGACGA3′

Lengthener (polyA) and enzyme terminator (iSp9) modifications of the antisense primer resulted in a longer length of the antisense strands[35] allowing for size separation and recovery of the target biotinylated sense ssDNA strand by gel excision from 4% denaturing agarose E-gels with final purification by gel extraction columns (Nucleospin, Macherey-Nagel). Biotinylated sense library strand was used for enrichment. Asymmetric PCR mixture (100 μl) contained 5× Q5 PCR buffer, 0.2 mM dNTPs, 0.08 μM of antisense primer, 30 μM of sense primer, 0.01 pmol template (of pure library) or 57 μl of post-dissection solution containing library/tissue (after enrichment) and 2 U of Q5 Hot Start High-Fidelity DNA polymerase. PCR programme included initial denaturation at 98 °C for 30 s, followed by cycle of denaturation, annealing (60 °C for 30 s) and extension (72 °C for 3 min), and final extension was at 72 °C for 5 min. For pure library, 15 cycles of amplification were performed; for libraries during enrichment number of cycles varies between 15 and 30 depends on the recovery. Asymmetric PCR products were mixed with denaturing buffer (180 mM NaOH, 6 mM EDTA), heated at 70 °C for 10 min, cooled down on ice for 3 min, loaded ~20 μl on 4% agarose SYBRGold gel (E-GEL EX Gels, G401004, Life Technologies) and separated for 15 min. Single-stranded sense strand was cut, gel pieces were combined with NTC buffer (Nucleospin, Macherey-Nagel), melted at 50 °C for 5–10 min until all pieces got molten. In all, 700 μl of melted agarose was loaded onto Nucleospin column and then followed standard procedure for ssDNA purification. Purified DNA was eluted with 30 μl of NE buffer.

**FFPE tissue slide-based SELEX.** Enrichment of ssDNA libraries toward C+T treatment response was performed according to the scheme in Fig. 2. Treatment regimens for enrichment cases can be found in Supplementary Data 1. In the enrichment of each library, first three rounds were performed on positive cases only, followed by additional three rounds with two counter-selection cases and one positive case. For positive selection, 400 μl of blocking buffer (0.8 ng/μl sheared salmon sperm DNA (Life Technologies), 0.8 ng/μl tRNA (Life Technologies), 1 μg/μl HSA (Sigma), 0.5% F127 (Thermo Fisher) and 3 mM MgCl₂ in 1× phosphate-buffered saline (PBS)) was mixed with 90 μl of ssDNA library solution (7 pmol in 1× PBS, 3 mM MgCl₂) on top of the Agilent gasket slide. FFPE tissue slide, after deparaffinization and epitope retrieval, was mounted on top of the gasket slide containing binding cocktail and incubated for 1 h in Agilent microarray hybridization chambers with rotation at room temperature (RT). After incubation, slides were washed by dipping into 2 × 750 ml washing buffer (1× PBS, 3 mM MgCl₂), 3 dips into each jar. Next, 490 μl of NFR, supplemented with 3 mM MgCl₂, was added to the slide for 45 s and washed by 6 times dipping in 750 ml washing buffer. Based on the initial pathological diagnosis from the corresponding H&E slides, cancer areas were dissected and transferred into 180 μl water, which served as a template for asymmetric PCR with unequal length primers to generate single-stranded library for the next round (see protocol above). Remaining normal tissue served for internal counter-selection. This protocol was repeated for three rounds. For negative selection, binding cocktail was added directly to the tissue of counter-selection slides and incubated for 1 h in a humidity chamber. After incubation, the maximum volume of supernatant was collected. Additionally, 50 μl of blocking buffer was applied to collect the unbound ssDNAs. Combined supernatant was added to the second counter-selection slide and incubated for 1 h. After incubation, supernatant was collected the same way as before and applied to the slide from the positive case for another hour incubation, done this time in the Agilent microarray hybridization chamber. The following steps, washing, staining and PCR, were the same as described above.

**PLP using the enriched libraries.** Staining of FFPE tissue slides with enriched libraries was performed on Dako Autostainer. After baking slides at 60 °C for 1.5 h, epitope retrieval was done on Dako PT-Linker at pH9, 98 °C, 22 min. The staining on Dako autostainer includes 5 min peroxidase inhibition with 450 μl of solution, containing disodium hydrogenorthophosphate 5% ≤ x < 7%, H₂O₂ 3% ≤ x < 5%, phosphoric acid, monosodium salt, monohydrate 1% ≤ x < 2%, 1 h incubation with 450 μl of binding cocktail (3.4 pmol of enriched library, 0.65 ng/μl sheared salmon sperm DNA, 0.65 ng/μl yeast tRNA, 10% BlockAid (Life Technologies), 30 min incubation with 2 μg/ml of Streptavidin Poly-HRP (Life Technologies), supplemented with 3 mM MgCl₂, 10 min staining with DAB solution, supplemented with 3 mM MgCl₂, followed by 5 min incubation with 450 μl of Hematoxylin (2 ng/μl final conc.). Rinsing with 1× PBS, 3 mM MgCl₂ buffer was done between each step. Finally, the stained slides were dehydrated with ethanol and xylene and covered by coverslip for long-term storage. Microscopy was done on an Olympus BX41.

Histological scores for both nuclear and cytoplasmic staining were calculated as the sum between intensity levels (1, 2 and 3) multiplied by the percentage of the cells with this particular intensity.

**Next-generation sequencing.** We utilized the modified version of the two-step library preparation protocol from Illumina for sequencing low-diversity libraries[23].

**Equipment and settings.** Microscopy and scoring of PLP staining intensities was done on an Olympus BX51 using standard bright field settings. Microscopy and imaging of the PLP-stained slide was done on an Olympus BX41 using standard bright field settings, with ×4, ×10 and ×20 objectives (UPlanFL N). Image acquisition camera: Olympus DP25. Image acquisition software: OLYMPUS DP2-TWAIN, which imports images, captured by DP25, via the TWAIN_64 Twacker (version 2.0 12/29/2008, TWAIN Working Group). Acquisition settings: Exposure: automatic. Sensitivity: ISO 200. Exposure compensation: 1. Pseudo colour: off. API version: 3.0.38. Firmware version: 1.0.14. Driver version: 2.1.32.

*Image resolution data.* Dimensions: 2560 × 1920 pixels. Bit depth: 24. Xyzt: not applicable.

Imaging was done at RT. No further manipulations or adjustments were done to the images.

**Statistical analysis.** First, the ability of each single library to classify C+T or T treatment Bs and NBs was assessed by ROC curves and AUCs (Fig. 5a, b). To predict the patient's response to C+T therapy with a multivariable method utilizing PLP scores from both libraries EL-NB and EL-B staining, a logistic regression model was developed. Specifically, a binary outcome of responder/non-responder was used as the response variable and log-transformed and standardized staining scores of libraries EL-NB and EL-B were treated as independent variables (Fig. 5c, blue). A 10-fold CV was conducted to assess the generalizability of the model's prediction performance, in which the data set was randomly split into 10 equal parts exclusively. A logistic regression model was built on nine parts and subsequently tested on the one hold-out part. This process was iterated throughout the 10 parts, and only the predicted probability was collected for further assessment (Fig. 5c, black).

The endpoints of TTNT were defined as either the time of next non-trastuzumab treatment or death. Patients without the next non-trastuzumab treatment or death information were censored at the last contact date (see vertical marks in the KM curves (Fig. 6a, b). A Cox-PH model was fitted using either tumours' HER2 status or PLP test results as the independent variable. Median survival time was calculated from the KM estimate. Log-rank test was performed to evaluate the significance of TTNT survival between groups. All analysis was conducted using the "survival" r package.

Two-tailed *t*-test was performed for comparison of the groups of trastuzumab-treated patients with TTNT to those without TTNT data and to patients not treated with trastuzumab as shown in Supplementary Data 2.

**Data availability statement.** The authors declare that all data supporting the findings of this study are available within the paper and its Supplementary Figures and Data or available from the authors upon request.

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

## Acknowledgements

We thank Hunter Garner, Peggy Gates, Yvonne Veloso, Elma Contreras, Mike Nelson, James Castano, Liz Thomas, Joanna Macinskas, Annemarie Benton, Phil Ellis, Anthony Helmstetter and Inga Rose for help in sample collection and executing experiments and George D. Demetri for productive discussions.

## Author contributions

D.S., G.P., M.F., G.M., J.V., V.D. and D.D.H. developed the concept of poly-ligand profiling; M.F., G.M., D.S., V.D., Z.G., A.S., P.K., B.T., S.L., J.W., R.S. and X.W. designed and performed experiments; A.V., R.G., G.R. and S.P.G. provided patient data and tissue. D.A.B., J.Q., A.B., L.S.S., J.V., R.G., G.M., D.M., G.P., A.B.H., J.M., N.X., J.W., V.D., Z.G., M.M., A.V., G.V., D.S. and M.F. analysed and discussed the experimental results; M.F., G. M., D.S., V.D., M.M., J.W., D.A.B., A.B.H. and G.P. wrote the manuscript.
