## [Peer Review File(PDF 126 kb) · Nature Communications]

Reviewers' comments:

Reviewer #1 (Remarks to the Author):

Overall, the authors have adequately addressed many of the reviewers concerns.

With regards to my major concern (sequencing of the enriched libraries) this is necessary to identify a "true and consistent sequence signature" that will enable to identify responders from non-responders every single time.

In the aptamer (selex) field, we are well aware that every time we amplify a given enriched round, we get a slightly different sequence representation. This is due to the nature of the amplification step (insertion of mutations during amplification or biased amplification of sequences). If the authors want to have a consistent/reliable clinical test/assay to assess responders from non-responders they need to determine a priori the sequence signatures as the sequences in any given round will change upon rounds of amplification (even in the absence of selection - see Thiel et al. NAT 2011).

In my opinion this remains a critical point that needs to be raised in the manuscript (even if just discussed).

Reviewer #2 (Remarks to the Author):

The revised efforts of Domenyuk and colleagues is improved and clarified. They have addressed many, but not all of the key criticisms in my original review. There are 2 key issues that remain, listed below.

1. My initial concern regarding H-scoring and single pathologist interpretation was addressed by selection of a small subset for automated scoring (n=26). While this is a step in the right direction, I see no reason why they could not do the automated scoring on the whole cohort. The automated scoring relieves the authors of the need to engage more pathologists, but does not excuse them to sample less than ½ their cohort for automated scoring.

2. The second key concern is the absence of a validation set from a uniformly treated population. While they show data from a trastuzumab treated and chemo only treated subsets, the data still represents relatively small numbers. They state that prospective validation is beyond the scope of the current study. I understand the pilot nature of the current study, but I still believe a validation set (even if retrospective) is required beyond the 120 cases shown, to make the work credible.

Reviewer 2 commented on reviewer 1 (at Nature) responses to the editor only.

This reviewer feels that comment 1 regarding about the relatively low number of cases cannot be addressed. Comment 2 and 3 are addressed satisfactorily. Finally regarding point 4 concerning the nature of the measured phenomenon, the reviewer suggests that an interaction test might be necessary to claim predictive value. The minor concerns were sufficiently addressed.

Referee #1 (Remarks to the Author):

Reviewer #1 found that overall, we have adequately addressed many of the reviewers concerns but still had a remaining issue regarding the sequencing of the libraries.

1. With regards to my major concern (sequencing of the enriched libraries) this is necessary to identify a "true and consistent sequence signature" that will enable to identify responders from non-responders every single time. In the aptamer (selex) field, we are well aware that every time we amplify a given enriched round, we get a slightly different sequence representation. This is due to the nature of the amplification step (insertion of mutations during amplification or biased amplification of sequences). If the authors want to have a consistent/reliable clinical test/assay to assess responders from non-responders they need to determine a priori the sequence signatures as the sequences in any given round will change upon rounds of amplification (even in the absence of selection - see Thiel et al. NAT 2011). In my opinion this remains a critical point that needs to be raised in the manuscript (even if just discussed).

Answer: Thiel et al show in the paper mentioned by Reviewer#1 that PCR can affect the composition of libraries and we understand that it is important to analyze the effect of library amplification by PCR, especially if the libraries will be part of a clinical test. We show in Extended Data Figure 4 (now 4A) that up to 10 generations of PCR amplification of EL-NB and EL-B do not significantly affect the staining intensities and profiles of the libraries. In response to Reviewer#1 we have now added data sets that analyze the sequence composition of TL-NB and TL-B after subsequent generations of PCR amplification (new Extended Data Figure 4B), and have added Thiel et al. (Ref 32). The plots show the percent overlap between top sequences (at different cut-offs) in generation 1 and the following 9 generations. Please note, only the perfect match between sequences was allowed in this analysis. As expected, in the later generations there is a declining trend of perfectly matching sequences, which is most likely caused by point mutants of the original sequences, slowly increasing in their representation at later generations in the library. As shown in the Extended Data Fig. 4A, such changes in library composition did not compromise the staining intensity and profile. Furthermore, there is no change in the first 8 generations of library TL-B and in the first 6 generations of library TL-NB. In fact, if we will amplify all amount of library by generation 4 we will get enough for $>10^9$ tests. Accordingly, there is no practical reason to explore later generations. These results are now presented in Extended Data Fig. 4B and the related discussion was added to the manuscript on p. 7, last §. Inclusion of these data necessitated adding Daniel Magee as another author.

Regarding the the informative "sequence signature": NGS is not ideal for this representation, because NGS data are based on the total recovery of the library from the slides, without split into groups of sequences specifically staining nucleus and/or cytoplasm compartments of cancer or stroma cells. Thus, NGS does allow to see what sequences are driving staining overall but does not allow to decipher their inter-cellular type or sub-cellular compartments specificity.

Reviewer #2 (Remarks to the Author):

Reviewer #2 commented that our revised efforts is improved and clarified and that we have addressed many, but not all of the key criticisms in his/her original review. Two key issues remained:

1. My initial concern regarding H-scoring and single pathologist interpretation was addressed by selection of a small subset for automated scoring (n=26). While this is a step in the right direction, I see no reason why they could not do the automated scoring on the whole cohort. The

automated scoring relieves the authors of the need to engage more pathologists, but does not excuse them to sample less than ½ their cohort for automated scoring.

Answer: We agree with Reviewer#2 and have scanned all test set slides (n = 61), using the Coreo scanner from Ventana, and performed autoscoring analysis, using the Visiopharm software as before. The resulting ROC curve is based on the combined scores from EL-NB and EL-B libraries and is shown below:

2. The second key concern is the absence of a validation set from a uniformly treated population. While they show data from a trastuzumab treated and chemo only treated subsets, the data still represents relatively small numbers. They state that prospective validation is beyond the scope of the current study. I understand the pilot nature of the current study, but I still believe a validation set (even if retrospective) is required beyond the 120 cases shown, to make the work credible.

Answer: While Reviewer #2 acknowledges (also when commenting on reviewer #1 from Nature) that a prospective validation is beyond the scope of this manuscript, the reviewer prefers to see a validation set (even if retrospective) beyond the 61 trastuzumab-treated cases plus 63 Pt/taxane-treated cases shown. We would love to address this point. However, despite many attempts over a protracted period of time, we have been unable to acquire samples from patients with homogeneous treatment histories and suitable outcome data. Indeed, we are facing a catch 22 situation; we consistently hear from investigators and sponsors that they require a peer-reviewed publication to convince them to relinquish their precious samples. At the same time the reviewer asks for additional samples to support publication. While we acknowledge that larger data sets are always preferred, we would like to reiterate that our study did in fact include an independent, multi-institutional cohort to validate the efficacy of the technology. It took a long period of time and substantial financial investment to acquire the 124 cases in our study and we have simply exhausted all avenues and resources available to obtain an additional set.

Reviewer#2 comments to reviewer#1 from Nature:

This reviewer feels that comment 1 regarding about the relatively low number of cases cannot be addressed. Comment 2 and 3 are addressed satisfactorily. Finally regarding point 4 concerning the nature of the measured phenomenon, the reviewer suggests that an interaction test might be necessary to claim predictive value. The minor concerns were sufficiently addressed.

Answer: Regarding the nature of the measured phenomenon (interaction test): We analyzed and compared the key clinical and demographic parameters of the B/NB groups in both Trastuzumab and Platinum/Taxane treated cohorts using analysis of variance (ANOVA). Results are added to the Supplementary Table 4 (see the 2nd tab "BvsNB in Trast vs Plat-Tax"). No significant difference was observed between mentioned parameters in all groups. This result shows that there is no evidence that the detected difference in response to trastuzumab is influenced by the listed clinical/demographic parameters.

We are grateful to the reviewers for their comments and the time spent on evaluating our manuscript. We found their input to be highly valuable.

Sincerely,

Michael Famulok, Günter Mayer, and David Spetzler

REVIEWERS' COMMENTS:

Reviewer #1 (Remarks to the Author):

The authors have adequately addressed this reviewer's concerns

Reviewer #2 (Remarks to the Author):

I had two points remaining to be addressed. The authors responded to the first by scanning the remaining cases, but could not respond to the second point related adding a validation set.

No further comments